# Postnatal Choline Supplementation Rescues Deficits in Synaptic Plasticity Following Prenatal Ethanol Exposure

**DOI:** 10.3390/nu14102004

**Published:** 2022-05-10

**Authors:** Erin L. Grafe, Mira M. M. Wade, Claire E. Hodson, Jennifer D. Thomas, Brian R. Christie

**Affiliations:** 1Division of Medical Sciences, University of Victoria, Victoria, BC V8W 2Y2, Canada; egrafe@uvic.ca (E.L.G.); euromira15@gmail.com (M.M.M.W.); clairehodson1@gmail.com (C.E.H.); brain64@uvic.ca (B.R.C.); 2Department of Psychology, San Diego State University, San Diego, CA 92120, USA

**Keywords:** prenatal ethanol exposure, fetal alcohol, hippocampus, synaptic plasticity, choline supplementation, dentate gyrus, sex differences, intervention

## Abstract

Prenatal ethanol exposure (PNEE) is a leading cause of neurodevelopmental impairments, yet treatments for individuals with PNEE are limited. Importantly, postnatal supplementation with the essential nutrient choline can attenuate some adverse effects of PNEE on cognitive development; however, the mechanisms of action for choline supplementation remain unclear. This study used an animal model to determine if choline supplementation could restore hippocampal synaptic plasticity that is normally impaired by prenatal alcohol. Throughout gestation, pregnant Sprague Dawley rats were fed an ethanol liquid diet (35.5% ethanol-derived calories). Offspring were injected with choline chloride (100 mg/kg/day) from postnatal days (PD) 10–30, and then used for in vitro electrophysiology experiments as juveniles (PD 31–35). High-frequency conditioning stimuli were used to induce long-term potentiation (LTP) in the medial perforant path input to the dentate gyrus of the hippocampus. PNEE altered synaptic transmission in female offspring by increasing excitability, an effect that was mitigated with choline supplementation. In contrast, PNEE juvenile males had decreased LTP compared to controls, and this was rescued by choline supplementation. These data demonstrate sex-specific changes in plasticity following PNEE, and provide evidence that choline-related improvements in cognitive functioning may be due to its positive impact on hippocampal synaptic physiology.

## 1. Introduction

Fetal Alcohol Spectrum Disorders (FASD) is a term encompassing a range of conditions caused by exposure to alcohol during the prenatal period. Prenatal ethanol exposure (PNEE) results in impairments in executive, motor and emotional functioning, and learning and memory [1]. FASD affects between 1–5% of the population [2,3]; however, this may be an underestimation due to difficulties in diagnosis. Although the abstinence of alcohol consumption during pregnancy is the only way to prevent FASD, prominent social issues such as poor education around the risks of PNEE, domestic violence, or mental health disorders contribute to the estimated 10% of women who continue to drink during pregnancy [4,5]. Given these complex social issues, there is still a significant need to research potential therapies following PNEE. Although exercise [6,7,8,9], antioxidant supplementation [10,11], and pharmaceuticals [12] have been shown to improve some impairments associated with PNEE, currently, there is a paucity of treatment options used in clinical populations [12].

One promising therapy for FASD is choline supplementation. Choline is an essential nutrient found in foods such as liver, eggs, and wheat germ [13]. Choline and its metabolites are involved in many cellular processes necessary for proper tissue functioning, including methyl donation in the one-carbon metabolism cycle, phospholipid generation, and formation of the neurotransmitter acetylcholine [14]. The Institute of Medicine has set adequate choline intake to be 450 mg/day and 550 mg/day for pregnant and lactating persons, respectively [13]; however, most adults do not meet these recommendations [15]. In fact, poor perinatal choline intake has been raised as a public health concern by experts in the field [16,17]. However, many studies have demonstrated that sufficient or elevated prenatal choline supplementation improves cognitive scores [18,19] and outcomes following immune challenge [20]. The importance of developmental choline is evidenced through the significant elevation of serum choline in the neonatal rat [21] and the relative increase in permeability across the blood brain barrier [22]. Despite a reduction in the permeability of choline in the adult brain [22], choline can nonetheless cross the blood brain barrier effectively through choline transporter-like proteins (CTL-1 and CTL-2) to maintain appropriate levels of neuronal choline concentrations [23].

Choline supplementation during both the prenatal and postnatal period has the potential as a treatment for FASD. Although there is strong evidence regarding the cognitive benefits of prenatal choline supplementation during PNEE [24,25,26,27,28,29,30,31], a large proportion of individuals with FASD are not diagnosed until childhood. Therefore, there is a need for postnatal treatment programs targeting FASD. Postnatal choline supplementation has been shown to improve some cognitive measures following PNEE [24,32]. For example, 100 mg/kg/day supplementation of choline chloride administered from postnatal day (PD) 10–30 improved outcomes on the Morris Water Maze [33,34], a task that depends on the functional integrity of the hippocampus. Indeed, postnatal choline supplementation is already being tested in pediatric clinical populations diagnosed with FASD with minimal reported adverse effects, namely a fishy body odor [35,36]. Initial cognitive tests have demonstrated enhanced working memory and IQ at a 4-year follow-up [35,36]. However, in a separate study, choline administration in older children with FASD did not show significant changes in cognitive function, indicating there may be a limited developmental window for treatment efficacy [37]. These findings are promising, but many questions remain about the mechanism by which choline is acting in the ethanol-exposed brain to improve learning and memory and how long these benefits persist.

Deficits in hippocampal synaptic plasticity observed following PNEE [38,39,40,41] have mirrored the learning and memory impairments associated with FASD. These deficits can also be sex specific. In male offspring, PNEE reliably impairs long-term potentiation (LTP) and alters long-term depression (LTD), mechanisms of synaptic plasticity, throughout the lifespan [41,42,43,44,45,46]. However, studies that include female offspring with PNEE have demonstrated impaired bidirectional synaptic plasticity during the early juvenile period (PD 21–28) [42], but no effect or even increased LTP in later adolescence and into adulthood [46,47,48]. The reason for these sex-specific and age-dependent alterations with PNEE is unknown. Therefore, this study sought to answer how postnatal choline supplementation alters LTP in late juvenile PNEE male and female offspring. We hypothesized that PNEE male offspring would show deficits in LTP that would be mitigated with choline supplementation, whereas female offspring would be resilient to prenatal ethanol exposure.

## 2. Materials and Methods

### 2.1. Animals and Breeding

All procedures were performed following the University of Victoria Institutional Animal Care Committee and Canadian Council for Animal Care standards. Adult male and female Sprague Dawley rats (*Rattus norvegicus*; Charles River Laboratories, Quebec, QC, Canada) were housed in standard cages in colony rooms kept at 21° and maintained on a 12-h light–dark cycle. All animals had ad libitum access to drinking water and standard solid rat chow, except during the ethanol diet. Nulliparous adult female rats were paired with adult male rats overnight until detection of sperm in a vaginal smear. Gestational day (GD) 1 was labelled as the day of sperm detection. Females were immediately singly housed in standard conditions and randomly assigned into Control or Ethanol groups. All dams were weighed on GD 1, 7, 14, and 21 and typically gave birth the evening of GD 22.

### 2.2. Prenatal Diet Conditions

Animals in the control group received ad libitum access to solid rat chow (Pico Rodent Diet 5053 (irradiated) or Lab Diet Rodent 5001; 1575–1840 ppm choline) throughout gestation. Animals in the ethanol group received ad libitum access to a nutritionally-fortified liquid diet (Weinberg/Keiver high-protein liquid diet–experimental; Dyets Inc. (Bethlehem, PA, USA), No. 710324; 0.66 kg/L choline bitrate) containing 35.5% ethanol-derived calories. Dams were gradually exposed to the diet from GD 1–3 and remained on the ethanol diet until GD 21. The liquid diet was replaced approximately 2 h prior to the onset of the dark phase every day. From GD 22 onwards, dams were returned to the solid rat chow diet and typically gave birth on the evening of GD 22. This moderate ethanol exposure paradigm typically produces blood alcohol concentrations between 80–180 mg/dl [9,10,11,49]. See Figure 1A for the experimental timeline and Table 1 for litter demographics.

### 2.3. Postnatal Choline Supplementation

Litters were culled on postnatal day (PD) 2 to 12 pups (6 males and 6 females, when possible) and offspring were weighed on PD 2, 4, and then daily throughout the injection period. Offspring were randomly assigned to the choline-treated or saline control group on PD 10. Offspring received either a subcutaneous injection of choline chloride (100 mg/kg/day) or a volume-matched saline control solution (0.9% NaCl) from PD 10–30. This dose of choline has previously been demonstrated to ameliorate behavioral deficits in offspring exposed to ethanol in the third trimester equivalent period [34]. Offspring were weaned into sex-matched cages on PD 22. Of note, non-handled control animals were included in the control groups as there was no significant difference in the magnitude of LTP between the two groups (*n* = 6 slices control + saline males, *p* > 0.05; *n* = 6 slices control + saline females, *p* > 0.05).

### 2.4. In Vitro Electrophysiology

Male and female offspring were used for in vitro electrophysiology experiments from PD 31–35, as described previously [42]. Briefly, offspring were anesthetized with inhaled isoflurane and decapitated. Brains were rapidly extracted, and transverse slices (400 µm) were cut in ice-cold artificial cerebral spinal fluid (aCSF) composed of (in mM) 125 NaCl, 2.5 KCl, 1.25 NaH_2_PO_4_, 25 NaHCO_3_, 2 CaCl_2_, 1.3 MgCl_2_, and 1.4 dextrose bubbled continuously with carbogen (95% O_2_/5% CO_2_; 295–305 mOsm; pH 7.2). The slices were allowed to recover in warm aCSF (32 °C) for approximately 15 min before being maintained at room temperature (25 °C). At this point, the slices were left to recover for one hour before being used for electrophysiology at 32 °C.

The slices were visualized using an upright microscope (Olympus, BX50WI, Olympus, Center Valley, PA, USA) allowing a bipolar stimulating electrode (FHC, Bowdoinham, ME, USA) and glass recording micropipette filled with aCSF to be placed in the medial perforant pathway (MPP) of the dentate gyrus (DG) of the hippocampus. Field excitatory postsynaptic potentials (fEPSPs) were evoked with bipolar electrodes and square wave pulses (0.12 ms) and recorded using an Axon Multiclamp 700 B amplifier, digitized by an Axon Digidata 1440, and recorded using Clampex 10.5 software (Molecular Devices, San Jose, CA, USA). fEPSP placement was optimized such that fEPSP amplitude was a minimum of 0.7 mV and then set to 50% of the maximum. Paired pulse tests were conducted before high-frequency stimulation (HFS), consisting of 6 pairs of pulses 50 ms apart. Paired pulse ratios were calculated as the slope of pulse 2/slope of pulse 1. Input–output curves were generated by increasing pulse width from 30 to 300 µs and analyzing the fiber volley and the fEPSP amplitude with increasing pulse width.

Pre-conditioning recordings (1 pulse every 15 s; 0.067 Hz) were conducted in picrotoxin (Tocris, 100 µM) to block GABA_A_ receptor transmission until a stable baseline was achieved for 20 min (<10% change; <0.5 change in slope over 20 min). Then, a high-frequency stimulation protocol was delivered (HFS; 50 pulses @ 100 Hz, repeated four times with a 30 s intertrain interval), followed by 60 min of post-conditioning recording (1 pulse every 15 s) in regular aCSF. The slope of the fEPSP was analyzed, binned to 1 min intervals, and post-conditioning recordings were calculated as a percent change from the pre-conditioning average slope. Inclusion criteria included showing a minimum of 10% long-term potentiation and having a stable post-conditioning response from 50–60 min post-HFS (slope < 1.5).

### 2.5. Statistics & Analysis

Data were collected with pClamp 10.5 and Clampfit 10.5 (Axon Instruments, Molecular Devices, San Jose, CA, USA, RRID: SCR_011323). All data are presented as the mean ± standard error (SEM). Two-way ANOVAs were followed by Tukey post hoc analysis. When the Shapiro–Wilk test of normality was significant (*p* < 0.05), a logarithmic transformation of the data was used for analyses. I/O curves and weight data were analyzed using repeated measures ANOVA followed by Holm’s method for multiple comparisons. When Mauchly’s test of sphericity was significant (*p* < 0.05), the Greenhouse–Geisser correction was used. In these cases, the degrees of freedom are corrected, and the effect sizes are presented as partial eta squared (η_p_^2^).

## 3. Results

### 3.1. Reduced Body Weight following PNEE

All offspring gained weight throughout the injection period (*p* < 0.001). However, weight gain was significantly less across the injection period for both males (Figure 1B, F (2.61, 62.63) = 8.04, *p* < 0.01, η_p_^2^ = 0.25) and females (Figure 1C, F (2.10, 65.18) = 5.29, *p* < 0.01, η_p_^2^ = 0.15) following PNEE. Further analysis determined that body weights of PNEE offspring deviated from control offspring beginning at PD 26 for males (*p* < 0.05) and PD 29 for females (*p* < 0.05), up through PD 30. Choline supplementation did not alter body weight trajectories in PNEE or control animals of either sex (*p* > 0.05).

### 3.2. Basal Changes in Excitability Evident in Female PNEE Offspring

Paired pulse ratios (PPRs) are a metric to explore short-term plasticity at the presynaptic terminus and are elicited by delivering two pulses in quick succession. PPRs can demonstrate either facilitation, due to an increase in residual calcium after the first pulse, or depression, due to neurotransmitter depletion. Historically, both facilitation and depression have been demonstrated within the MPP [50,51]. The PPRs in both male (Figure 2A) and female offspring (Figure 2B) demonstrated subtle paired pulse facilitation. There was no effect of prenatal ethanol exposure (Male: F (1, 33) = 0.77, *p* = 0.388, η_p_^2^ = 0.02; Female: F (1, 33) = 0.02, *p* = 0.895, η_p_^2^ = 0.00) or postnatal choline treatment (Male: F (1, 33) = 0.25, *p* = 0.618, η_p_^2^ = 0.01; Female: F (1, 33) = 0.05, *p* = 0.488, η_p_^2^ = 0.02). These data therefore suggest potential alterations in plasticity are not due to changes in presynaptic neurotransmitter release mechanisms.

Input–output curves were used to examine changes in postsynaptic excitability between conditions and sexes. Both male and female offspring fEPSPs responded to increases in the pulse width with a larger fiber volley and evoked fEPSP amplitude (*p* < 0.001). Fiber volley amplitude was analyzed to determine relative axonal input across conditions. In male offspring (Figure 3A) there was no main effect of prenatal condition (F (1, 39) = 1.21, *p* = 0.278, η_p_^2^ = 0.03) or postnatal treatment (F (1, 39) = 1.15, *p* = 0.290, η_p_^2^ = 0.3), but there was a significant interaction between PNEE and choline treatment (F (1, 39) = 6.20, *p* = 0.017, η_p_^2^ = 0.14). Specifically, choline-supplemented PNEE males had larger fiber volley amplitudes than saline-treated PNEE males (*p* = 0.048). When pulse width was considered as a factor, increases in fiber volley were evident at pulse widths of 270 µs (*p* = 0.042) and 300 µs (*p* = 0.029). However, when examining fEPSP amplitude across conditions (Figure 3B), there was no effect due to prenatal condition (F (1, 42) = 0.07, *p* = 0.791, η_p_^2^ = 0.00) or postnatal treatment (F (1, 42) = 0.85 *p* = 0.360, η_p_^2^ = 0.02).

In female PNEE offspring (Figure 3C), there was no significant increase in fiber volley amplitude compared to control offspring (F (1, 40) = 3.08, *p* = 0.087, η_p_^2^ = 0.07) or with choline treatment (F (1, 40) = 0.43, *p* = 0.518, η_p_^2^ = 0.01). Subsequent analysis of fEPSP amplitude determined that there was a notable, albeit non-significant, interaction between prenatal condition and postnatal treatment across pulse width (F (1, 42) =4.03, *p* = 0.051, η_p_^2^ = 0.09), with seemingly larger fEPSP amplitude exhibited in saline-treated PNEE females as compared to choline-treated PNEE females. When pulse width was examined as a factor (F (1.44, 60.32) = 6.21, *p* = 0.008, η_p_^2^ = 0.13), post hoc analysis determined this difference to occur between saline- and choline-treated female PNEE offspring at 240 µs (*p* = 0.044), 270 µs (*p* = 0.025), and 300 µs (*p* = 0.020). Of note, the average fEPSP amplitude during the preconditioning recording did not differ between conditions for males (C + S: −0.70 ± 0.07 mV, C + C: -0.65 ± 0.08 mV, P + S: −0.77 ± 0.11 mV, P + C: −0.66 ± 0.04 mV) and females (C + S: −0.67 ± 0.06 mV, C + C: −0.65 ± 0.04 mV, P + S: −0.75 ± 0.07 mV, P + C: −0.67 ± 0.05 mV).

### 3.3. Postnatal Choline Supplementation Ameliorated Deficits in PNEE Male Offspring LTP

High-frequency stimulation (HFS) was applied to elicit LTP in the dentate gyrus of juvenile offspring (Figure 4A, Table 2; *n* = 10–14 slices per group). Post-tetanic potentiation (PTP) was classified as the four responses acquired in the first minute following the conditioning stimuli. A decreased magnitude of PTP was apparent in male PNEE offspring (Figure 4B) as compared to Control offspring (F (1, 46) = 4.35, *p* = 0.043, η_p_^2^ = 0.09). Postnatal choline treatment, however, did not significantly alter ethanol’s effects on PTP magnitudes (F (1, 46) = 2.90, *p* = 0.096, η_p_^2^ = 0.06). Subsequently, we examined changes in LTP (minutes 50–60 following HFS; Figure 4C). There was no main effect of prenatal condition (F (1, 46) = 0.05, *p* = 0.829, η_p_^2^ = 0.00) or postnatal treatment (F (1, 46) = 3.12, *p* = 0.084, η_p_^2^ = 0.06); however, there was a significant interaction between prenatal condition and postnatal treatment (F (1, 46) = 4.17, *p* = 0.047, *p* = 0.022, η_p_^2^ = 0.11). Further *Tukey* post hoc analysis determined significant elevations in LTP in the PNEE + choline group compared to the PNEE + saline group (*p* = 0.039).

### 3.4. PNEE Increased LTP in Female Offspring

Female offspring have been reported to be resistant to the effects of PNEE on dentate gyrus synaptic plasticity in the adult and juvenile stages in vivo [46,48,52] and therefore we inquired as to whether these results would align with previous data. In female offspring (Figure 5A and Table 2; *n* = 11–14 slices per group), PTP was not affected by either PNEE (Figure 5B, F (1, 46) = 0.02, *p* = 0.888, η_p_^2^ = 0.00) or choline treatment (F (1, 46) = 0.34, *p* = 0.557, η_p_^2^ = 0.01). However, there was an increase in LTP with PNEE (Figure 5C, F (1, 46) = 4.20, *p* = 0.046, η_p_^2^ = 0.08). This effect was driven by an increase in LTP among PNEE subjects treated with choline, although there was no main effect of choline treatment on the magnitudes of LTP (F (1, 46) = 0.87, *p* = 0.356, η_p_^2^ = 0.05).

## 4. Discussion

This study is the first to examine how postnatal choline supplementation can impact synaptic plasticity following PNEE. We show that PNEE reduces LTP in the DG of male subjects and that postnatal choline supplementation can rescue this deficit. Conversely, while LTP was not reduced in females following PNEE, we observed hyperexcitability to increasing stimulation. This was ameliorated by postnatal choline treatment, indicating that, following PNEE, choline may help normalize excitability in females. Interestingly, there was no detectable effect of choline supplementation in the control groups. This could be related to the timing of choline supplementation, as most effects of choline in typically developing subjects are seen with prenatal, rather than postnatal, supplementation [53].

Interventions that include exercise and/or environmental enrichment [9,54,55,56,57,58,59], antioxidant supplementation [10,11,60], and other pharmaceutical therapies [61] have all shown some efficacy in mitigating ethanol’s effects. However, recent studies have demonstrated beneficial effects of the nutrient choline [29,62], and choline supplementation is emerging as one of the most promising treatments for FASD [32,62]. The present data illustrate that choline can mitigate the effects of PNEE on alterations in hippocampal plasticity. Decreased LTP has been consistently demonstrated in multiple hippocampus regions in males following PNEE [40]. This study is the first to show that postnatal choline supplementation can increase the magnitude of LTP in male juvenile PNEE offspring. Future studies will seek to elucidate the synaptic mechanisms behind these effects.

In contrast, female offspring appear resilient to the effects of PNEE on hippocampal synaptic plasticity [46,47,48]. This resiliency persists even following ovariectomy [46], suggesting that circulating hormones are not responsible for resilience to PNEE, although changes in local estrogen production in the brain have yet to be ruled out. Contrary to these findings, decreased bidirectional plasticity has been observed in younger (PD 21–28) PNEE female offspring [42]. Thus, there may be age-related compensation for an early-life ethanol insult within the female brain. Indeed, recent work has indicated a sex-specific alteration in inhibition across puberty in female offspring that increased the threshold for LTP induction [63]. This increase in inhibitory tone was related to a higher density of α5 subunit-containing GABA_A_ receptors within the adult female brain [63]. Within this context, it is essential to note that the conditioning stimuli in our recordings are performed in the presence of picrotoxin, a GABA_A_ receptor antagonist. Therefore, the hyperexcitable state in juvenile PNEE females could be due to increased inhibition usually masked by including picrotoxin in the aCSF. Rather, these data support the hypothesis that a compensatory increase in excitation could be utilized to overcome the threshold for LTP induction in juvenile PNEE female offspring, resulting in no overall decrease in LTP. To the best of our knowledge, this line of inquiry has not been studied, but may further our understanding of how the female brain adapts following PNEE.

Importantly, postnatal choline supplementation returned the female PNEE brain to control levels of excitability, but non-significantly increased the fiber volley amplitude. However, in PNEE male offspring, there was a significant increase in fiber volley amplitude with increasing pulse width among those treated with choline. The increase in fiber volley amplitude in relation to the fEPSP amplitude was surprising, as this did not impact the magnitude of LTP. Therefore, while the mechanism by which choline is acting is still unknown, these data indicate changes in synaptic transmission in the juvenile PNEE brain. Further experiments uncovering changes in the excitation–inhibition balance are thus needed to indicate the cause of altered synaptic transmission.

Choline could be modifying ethanol’s effects on plasticity via several mechanisms. Choline is involved in many cellular processes [14]; one potential area of interest is methylation changes in the epigenetic signature [64]. Indeed, studies in both individuals with FASD and animal models have demonstrated disruptions in the epigenetic signature with PNEE [65,66,67,68,69]. As choline is a methyl donor within the one-carbon metabolism cycle, it is perhaps unsurprising that postnatal choline supplementation has influenced methylation indices [31,70,71]. Whether there is a relationship between functional changes in synaptic transmission and methylation patterns will be an essential question to answer. Furthermore, choline supplementation could be altering cholinergic transmission. PNEE has been shown to alter hippocampal muscarinic receptor density [72] and reduce cholinergic neurons within the medial septum/diagonal band of the basal forebrain [73]. These neurons provide cholinergic input to the hippocampus. As acetylcholine acts as a neuromodulator of hippocampal function [74], increased cholinergic transmission could drive the circuit past LTP induction thresholds. It is also important to note that choline supplementation may impact peripheral locations as well, including the liver and muscle [14,16]. This supports the hypothesis that combined exercise and choline could have additive benefits following PNEE, perhaps by altering expression of brain derived neurotrophic factor (BDNF) [75,76].

Choline-related modifications in synaptic plasticity were evident shortly after choline treatment had ceased. Future work will need to investigate if improvements in hippocampal LTP due to choline supplementation continue into adulthood, and whether continued treatment throughout the lifespan is necessary to maintain the beneficial effects on brain functionality. Behavioral data have demonstrated that choline supplementation during the juvenile period (PD 10–30) improves performance on hippocampal-dependent behaviors into adulthood [33,34], whereas choline supplementation from PD 40–60 only improved behaviors related to the prefrontal cortex [77]. Whether changes in hippocampal synaptic plasticity are also maintained following choline treatment is yet to be explored. Nevertheless, there is promising evidence to suggest the maintenance of these positive effects, as Meck and colleagues (2008) determined that choline supplementation during two critical developmental time points, from GD 12–17 and PD 16–30, in the normally developing brain, caused improvements in spatial memory and increased hippocampal spine density in seven-month-old rats [53]. Furthermore, Pyapali and colleagues (1998) used a maternal choline supplementation paradigm from GD 12–17 without ethanol exposure and found long-lasting reductions in the threshold of LTP induction in 3–4 month-old rats [78]. Therefore, the effects of early life choline supplementation on synaptic plasticity may be long-lasting.

Currently, clinical trials are underway investigating the effects of choline supplementation in children with FASD [35,36,37]. When choline was supplemented for nine months in children aged 2.5–5 years old, improvements were demonstrated in learning and memory early on [36]. Many of these improvements were more robust in a 4-year follow-up, including significant improvements in working memory, IQ, and some learning and memory tests [37]. Thus, both the clinical data and preclinical literature support the assertion that early postnatal choline influences learning and memory. The data presented in this study support postnatal choline supplementation as a treatment for FASD and provide further insight into the neural mechanisms by which choline improves learning and memory outcomes.

## 5. Conclusions

This study demonstrates that postnatal choline supplementation exerts sex-specific benefits following prenatal ethanol exposure. Female PNEE offspring had divergent excitability, and male PNEE offspring had decreased LTP; choline supplementation returned both measures to control levels. This work provides a promising foundation for using postnatal choline supplementation as a treatment for PNEE and initiates further questions regarding the neural changes associated with improved behavioral outcomes.

## Figures and Tables

**Figure 1 nutrients-14-02004-f001:**
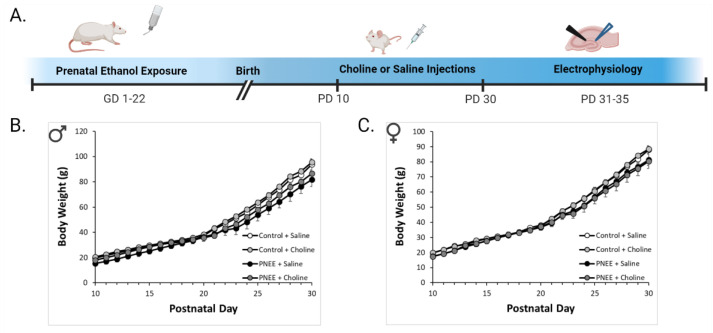
Choline treatment did not significantly affect body weight. (**A**). Cartoon outlining the experimental timeline for PNEE, choline administration, and electrophysiology. The average body weights (g) across the experimental timeline are shown for both (**B**) male offspring (*n* = 10–12 animals per group) and (**C**) female offspring (*n* = 6–14 animals per group).

**Figure 2 nutrients-14-02004-f002:**
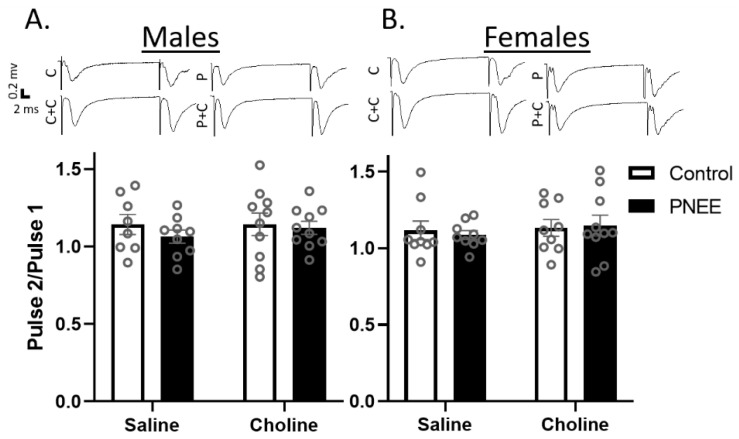
No changes in PPR with PNEE or choline treatment. The average paired pulse ratios (slope of pulse 2/pulse 1) are presented as bar graphs with individual data points as small circles for (**A**) male and (**B**) female offspring in all conditions. (C) Control + saline; (C + C) control + choline; (P) PNEE + saline; and (P + C) PNEE + choline. Representative paired pulse fEPSP traces are shown above each graph. Error bars are ±SEM.

**Figure 3 nutrients-14-02004-f003:**
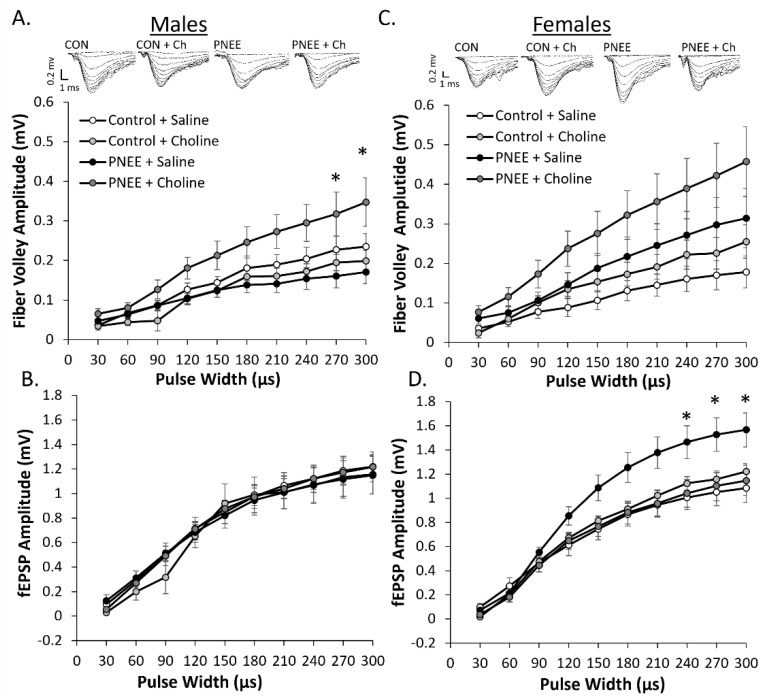
PNEE alters the stimulus response curve for field potentials in females. The average change in fiber volley amplitude with increasing pulse width for (**A**) male and (**B**) female offspring. The average change in fEPSP amplitude with increasing pulse width for (**C**) male and (**D**) female offspring. Control + saline = white, control + choline = light grey, PNEE + saline = black, and PNEE + choline = dark grey. Representative traces are above each graph. Error bars are ±SEM. * represents *p* < 0.05.

**Figure 4 nutrients-14-02004-f004:**
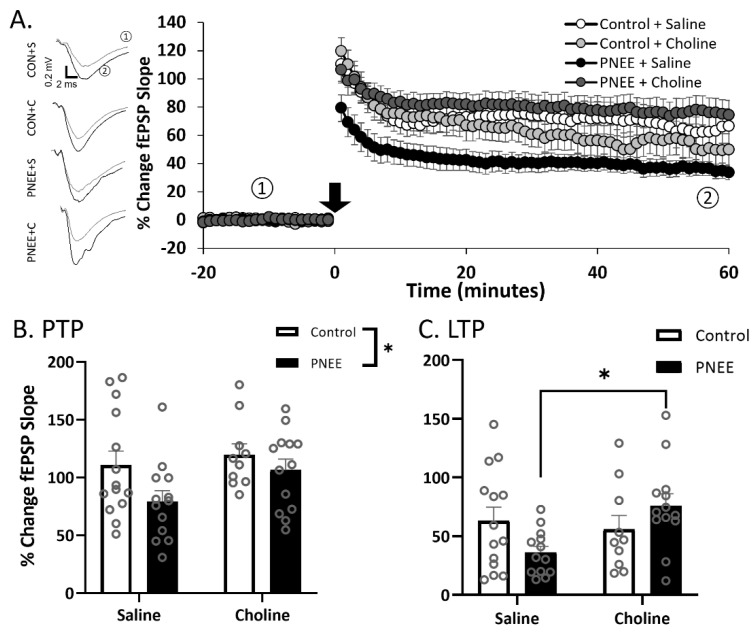
Decreased LTP in juvenile male offspring following PNEE is rescued with choline treatment. (**A**) Average LTP for each condition. Each point represents four traces binned to the one-minute interval. The black arrow represents the HFS conditioning stimulus. Representative traces are featured above for ① pre-conditioning and ② post-conditioning. Control + saline = white, control + choline = light grey, PNEE + saline = black, and PNEE + choline = dark grey. Average PTP ((**B**); first minute following HFS) and LTP ((**C**); minutes 50–60 following HFS) for each condition. * Represents *p* < 0.05. Error bars are ±SEM.

**Figure 5 nutrients-14-02004-f005:**
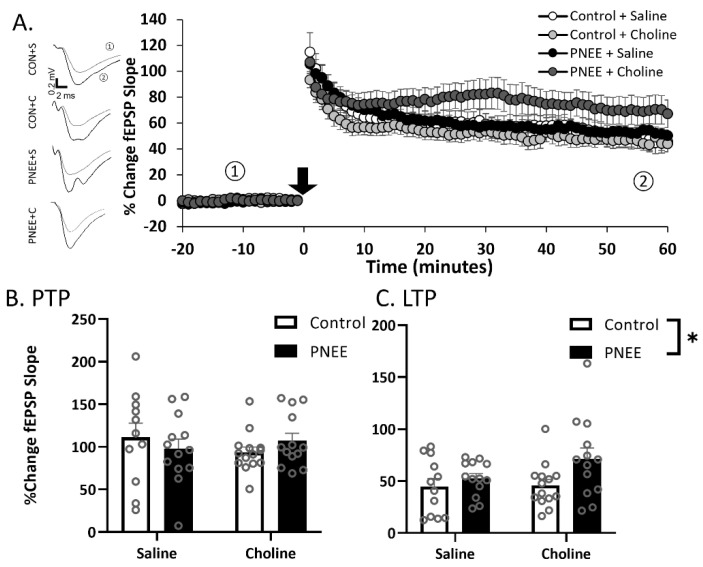
Increased LTP in juvenile female offspring following PNEE. (**A**) Average LTP for each condition. Each point represents four traces binned to the one-minute interval. The black arrow represents the HFS conditioning stimulus. Representative traces are featured above for ① pre-conditioning and ② post-conditioning. Control + saline = white, control + choline = light grey, PNEE + saline = black, and PNEE + choline = dark grey. Average PTP ((**B**); first minute following HFS) and LTP ((**C**); minutes 50–60 following HFS) for each condition. * Represents *p* < 0.05. Error bars are ±SEM.

**Table 1 nutrients-14-02004-t001:** Demographics of Control and PNEE Litters.

	Pup Sex Ratio (Male/Female)	Total Number of Pups	Gestation Length (Days)	Dam Weight (% Change)
Control (litters = 10)	1.0 ± 0.2	11.0 ± 0.5	21.5 ± 0.2	156.6 ± 7.9%
PNEE (litters = 13)	0.9 ± 0.1	9.3 ± 0.7	22.0 ± 0.1%	137.4 ± 4.7%

Average litter characteristics across treatment conditions ± SEM. A 2-tailed Student’s *t*-test was used to evaluate the effects due to PNEE. There was no significant effect of PNEE on pup sex ratio (*p* = 0.747), total number of live pups on PD2 (*p* = 0.083), gestational length (*p* = 0.076), or % change in dam weight (*p* = 0.075). All values are represented as ±SEM.

**Table 2 nutrients-14-02004-t002:** Average PTP and LTP for male and female juvenile offspring ± SEM.

	Males			Females		
	Slice (*n*), Animal (a), Litter (l)	PTP	LTP	Slice (*n*), Animal (a), Litter (l)	PTP	LTP
Control + Saline	*n* = 14, *a* = 6, *l* = 6	110.6 ± 12.3%	63 ± 11.5%	*n* = 11, *a* = 6, *l* = 5	114.9 ± 15.0%	48.3 ± 7.4%
Control + Choline	*n* = 10, *a* = 4, *l* = 3	119.6 ± 9.6%	52.7 ± 11.7%	*n* = 14, *a* = 5, *l* = 3	93.6 ± 6.3%	45.9 ± 5.8%
PNEE + Saline	*n* = 13, *a* = 6, *l* = 5	79.3 ± 9.4%	36.0 ± 5.4%	*n* = 12, *a* = 4, *l* = 4	97.8 ± 12.6%	53.0 ± 4.8%
PNEE + Choline	*n* = 13, *a* = 5, *l* = 5	106.5 ± 9.5%	76.0 ± 10.1%	*n* = 13, *a* = 4, *l* = 4	107.1 ± 8.8%	69.5 ± 10.9%

*n* = slice number, *a* = animal number, *l* = litter number. PNEE = prenatal ethanol exposure, PTP = post-tetanic potentiation, LTP = long-term potentiation. Each value represents average PTP or LTP ± SEM.

## Data Availability

The data presented in this study are available on request from the corresponding authors.

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
