# Peer review of "Postnatal Choline Supplementation Rescues Deficits in Synaptic Plasticity Following Prenatal Ethanol Exposure"

_nutrients, 2022, doi:10.3390/nu14102004_

Round 1
Reviewer 1 Report
This is interesting paper that uncovered the synaptic effects of choline supplementation in FASD model. Despite FASD led to different pattern of synaptic dysfunctions in male and female offspring, choline was beneficial for both. The latter suggests to action of the therapy on a root of the pathology.
I have only minor comments:
Can author add information in Introduction or Discussion:
It will be helpful to have some information about choline transport to the brain through BBB.
Also, injected choline, as pro-vitamin, can be uptaken by liver for storage and thereby not directly affect brain function.
The choline can act on peripheral ACh receptors in neuromuscular junctions (NMJs) and NMJs are main site where ACh used as neurotransmitter. It can be discussed. Since exercise, engaging neuromuscular transmission, might improve FASD.
Generally, there are two hypotheses that choline can directly act in brain or peripheral sites (liver, muscles), which can send signals to brain. I did not see this in introduction as well as the specific mechanisms underlying choline therapeutic action specifically in FASD.
What do the authors have to say about the side effects of choline?
Methods.
Why was this dose and route of introduction for choline chosen? Can author use lower and higher dose to study the key parameters?
I suggest to present data as Mean +_ SD, since standard error does not reflect the dispersion of data. Please see, for example, PMID: 25354300
Also, if data were not drawn from a normal distribution, then non-parametric tests should be used instead ANOVA. Do authors check normality of the distributions? It should be added to the Statistics.
Results.
Both control and choline groups of mice were injected many times. Does this procedure itself (in control) influence on detected parameters? Can authors compare control injected group vs intact mice?
It will be helpful if choline levels in brain as well as liver and skeletal muscles were estimated in PNEE vs control groups and effects of choline treatment on this were assessed.
Fig.2. As I can understand there are pared pulse facilitation and depression depending on inter-pulse intervals. Since to test short-term plasticity different intervals should be tested to have a possibility to judge about this type of plasticity. I cannot catch why authors used only one time in this case (and under these conditions the ratio of P2/P1 in control and PNEE groups was similar and close to 1).
Some behavioral tests can improve an underhanging of the relevance of choline-treatment induced synaptic changes in global brain functionality.
Author Response
We appreciate the efforts of the reviewer in providing insightful comments and helping to improve our manuscript. Our response to their queries are below.
Reviewer 2:
- “It will be helpful to have some information about choline transport to the brain through BBB.”
Response: We agree that this could be helpful and have added in relevant information to the introduction (lines 54-59)
- “Also, injected choline, as pro-vitamin, can be uptaken by liver for storage and thereby not directly affect brain function. The choline can act on peripheral ACh receptors in neuromuscular junctions (NMJs) and NMJs are main site where ACh used as neurotransmitter. It can be discussed. Since exercise, engaging neuromuscular transmission, might improve FASD.”
Response: This is an interesting addition to the potential mechanisms of choline. We have added a section within the discussion mentioning exercise and choline as a combined treatment (lines 364-369).
- “Generally, there are two hypotheses that choline can directly act in brain or peripheral sites (liver, muscles), which can send signals to brain. I did not see this in introduction as well as the specific mechanisms underlying choline therapeutic action specifically in FASD.”
Response: Similar to the response above, we have now added in a couple of sentences within the discussion to mention the potential peripheral impacts of choline supplementation. This would be an interesting avenue to explore, however we feel this is beyond the scope of the current study.
- “What do the authors have to say about the side effects of choline?”
Response: We have added in a note in the introduction regarding the minimal adverse effects of choline supplementation reported in clinical studies (lines 68-70). With this animal model and dose of choline, we find no noticeable side effects.
- “Why was this dose and route of introduction for choline chosen? Can author use lower and higher dose to study the key parameters?”
- Response: Previous work (Thomas et al., 2007) has administered a range of choline doses, from 10 mg/kg to 100 mg/kg. Doses as low as 10 mg/kg have been effective in improving learning deficits, however higher doses (50-100 mg/kg) mitigated ethanol-related hyperactivity as well. Therefore, while we agree that a dose response would be an interesting addition, we wanted to demonstrate positive effects at a known effective dose first. Similarly, other routes of choline have been examined as well, including intubation, which also effectively mitigates the effects of prenatal alcohol exposure on cognitive function. However, injections have been found to be effective and less stressful.
- I suggest to present data as Mean +_ SD, since standard error does not reflect the dispersion of data. Please see, for example, PMID: 25354300
Response: We appreciate this suggestion, but since we are showing individual data points in many of our graphs, the data dispersion is clearly presented. Thus, we have opted to maintain SEM within our graphs.
- “Also, if data were not drawn from a normal distribution, then non-parametric tests should be used instead ANOVA. Do authors check normality of the distributions? It should be added to the Statistics.”
Response: We thank the reviewer for catching this oversight. Normality was assessed using the Shapiro-Wilk test and in two instances did not hold (LTP measurements for both males and females). In these instances, we decided to transform the data using a logarithmic transformation. This decision was made in order to still assess the interaction between prenatal condition and postnatal treatment, which would not be possible using the Kruskal-Wallis non-parametric test. Importantly, analyzing the data this way did not change any of our findings.
- “Both control and choline groups of mice were injected many times. Does this procedure itself (in control) influence on detected parameters? Can authors compare control injected group vs intact mice?”
Response: A subset of control rats were not injected to determine the effect of repeated handling/injections on LTP. There was no significant difference in the amount of LTP in control male or female offspring so these groups (non-handled and injected) were combined and noted in the methods section.
- “It will be helpful if choline levels in brain as well as liver and skeletal muscles were estimated in PNEE vs control groups and effects of choline treatment on this were assessed.”
Response: Previously, we have not found that PNEE changes choline levels in the fetus or offspring brain or liver, although there may be transient changes in metabolites during intoxication. We have not examined levels in skeletal muscles. We agree that this would be an interesting to investigate and will consider in future studies, although this would be outside the scope of the present study.
- “Fig.2. As I can understand there are pared pulse facilitation and depression depending on inter-pulse intervals. Since to test short-term plasticity different intervals should be tested to have a possibility to judge about this type of plasticity. I cannot catch why authors used only one time in this case (and under these conditions the ratio of P2/P1 in control and PNEE groups was similar and close to 1).”
Response: While we agree that different intertrain intervals (ITI) can be utilized to determine changes in short term plasticity across of spectrum of protocols, it is however common to utilize one paired-pulse test between 50-100 ms as an indirect method to assess changes in presynaptic transmitter release. This time interval is preferred because ITIs shorter than this are contaminated by the initial pulse, whereas longer ITIs allow sufficient time for presynaptic calcium to disperse. Thus, the ITI used in these studies reflects best practice for this brain region and has been used in several previous publications by our laboratory.
- “Some behavioral tests can improve an underhanging of the relevance of choline-treatment induced synaptic changes in global brain functionality.”
Response: Behavioural tests would demonstrate the translational relevance of choline treatment on hippocampal plasticity. However, previous work from the Thomas laboratory has already demonstrated behavioural benefits of choline treatment on numerous tasks when administered either prenatally (with prenatal alcohol) or postnatally (following either prenatal or third-trimester equivalent alcohol exposure). Specifically, choline supplementation (100 mg/kg) from PD 10-30 has been shown to improve performance on numerous tasks that depend on hippocampal function in subjects exposed to developmental alcohol. Behaviors were not included in the present study, as they could influence LTP. Thus, the present study looks at baseline changes in plasticity. Studies are currently underway to examine how choline may modify the effectiveness of other treatments (i.e. exercise) for reducing FASD.
Reviewer 2 Report
In this study, the authors highlighted the role of postnatal choline supplementation in enhancing the negative effect of prenatal ethanol exposure (PNEE) on cognitive development. In particular, they analyzed hippocampal synaptic plasticity in the animal model (rats) after treatment with choline in both the control and alcohol-exposed groups.
The manuscript is generally well written and only few changes are needed. In particular:
- As the fetus is particularly sensitive to the mother's diet during pregnancy, It would be necessary to indicate the content of the control diet.
- An accurate revision of typing errors is recommended.
Author Response
We appreciate the efforts of the reviewer in providing insightful comments and helping to improve our manuscript. Our response to their queries are below.
Reviewer 1:
- “As the fetus is particularly sensitive to the mother's diet during pregnancy, It would be necessary to indicate the content of the control diet.”
Response: We have added in the brand name of the solid rat chow to provide more transparency on the content of the control diet.
- “An accurate revision of typing errors is recommended.”
Response: we have reviewed the document thoroughly to correct any missed typing errors.